# Association between Bone Mineral Density and Oral Frailty on Renal Function: Findings from the Shika Study

**DOI:** 10.3390/healthcare11030314

**Published:** 2023-01-20

**Authors:** Shingo Nakai, Fumihiko Suzuki, Shigefumi Okamoto, Sakae Miyagi, Hiromasa Tsujiguchi, Akinori Hara, Thao Thi Thu Nguyen, Yukari Shimizu, Koichiro Hayashi, Keita Suzuki, Tomoko Kasahara, Masaharu Nakamura, Chie Takazawa, Takayuki Kannon, Atsushi Tajima, Hirohito Tsuboi, Noriyoshi Ogino, Tadashi Konoshita, Toshinari Takamura, Hiroyuki Nakamura

**Affiliations:** 1Department of Hygiene and Public Health, Faculty of Medicine, Institute of Medical, Pharmaceutical and Health Sciences, Kanazawa University, 13-1 Takaramachi, Kanazawa 920-8640, Japan; 2Department of Public Health, Graduate School of Advanced Preventive Medical Sciences, Kanazawa University, Kanazawa 920-8640, Japan; 3Community Medicine Support Dentistry, Ohu University Hospital, Koriyama 963-8611, Japan; 4Advanced Health Care Science Research Unit, Innovative Integrated Bio-Research Core, Institute for Frontier Science Initiative, Kanazawa University, Kanazawa 920-0942, Japan; 5Department of Clinical Laboratory Sciences, Faculty of Health Sciences, Institute of Medical, Pharmaceutical, and Health Sciences, Kanazawa University, Kanazawa 920-0942, Japan; 6Innovative Clinical Research Center, Kanazawa University, 13-1 Takaramachi, Kanazawa 920-8641, Japan; 7Advanced Preventive Medical Sciences Research Center, Kanazawa University, 1-13 Takaramachi, Kanazawa 920-8640, Japan; 8Faculty of Public Health, Haiphong University of Medicine and Pharmacy, Ngo Quyen, Hai Phong 180000, Vietnam; 9Faculty of Health Sciences, Department of Nursing, Komatsu University, 14-1 Mukaimotorimachi, Komatsu 923-0961, Japan; 10Department of Bioinformatics and Genomics, Graduate School of Advanced Preventive Medical Sciences, Kanazawa University, 13-1 Takaramachi, Kanazawa 920-8640, Japan; 11Graduate School of Human Nursing, The University of Shiga Prefecture, 2500 Hassaka-cho, Hikone 522-8533, Japan; 12Department of Environmental Medicine, Faculty of Medicine, Kochi University, Kohasu, Oko-cho, Nankoku 783-8505, Japan; 13Third Department of Internal Medicine, School of Medicine, University of Occupational and Environmental Health, Iseigaoka 1-1, Yahatanishi-ku, Kitakyushu 807-8555, Japan; 14Third Department of Internal Medicine, University of Fukui Faculty of Medical Sciences, 23-3 Matsuoka Shimoaizuki, Eiheiji-cho, Yoshida-gun, Fukui 910-1193, Japan; 15Department of Endocrinology and Metabolism, Kanazawa University Graduate School of Medical Sciences, Kanazawa 920-8640, Japan

**Keywords:** estimated glomerular filtration rate, logistic models, oral frailty, osteo-sono assessment index

## Abstract

The association between oral frailty (OFr) and body action has been investigated, but its association with systemic function remains unclear. Therefore, this cross-sectional study examined the association between OFr with decreased bone mineral density (BMD) and renal function in residents of Shika town, Ishikawa Prefecture, Japan aged ≥40 years. This study included 400 inhabitants. The OFr total score was assessed using three oral domains in the Kihon Checklist (a self-reported comprehensive health checklist), the number of teeth, and brushing frequency per day. Measurements were the estimated glomerular filtration rate (eGFR) and the osteo-sono assessment index (OSI). Using a two-way analysis of covariance (*p* = 0.002), significantly lower OSI was indicated in the eGFR < 60 and OFr group than in the eGFR of < 60 and non-OFr group after adjusting for age, body mass index, and drinking and smoking status as confounding factors. Multiple logistic regression analysis confirmed this relationship (*p* = 0.006). Therefore, lower BMD seems to be associated with lower renal function only when accompanied by OFr. Further longitudinal studies are needed to confirm these results.

## 1. Introduction

Oral frailty (OFr) is considered a poor oral condition or oral hypofunction [1,2,3,4]. A review by Azzolino et al. [1] indicated that insufficient oral health may influence food preference and nutrient intake, leading to malnutrition and, accordingly, to frailty and/or sarcopenia. A longitudinal study indicated that OFr approximately doubled the risk of systemic frailty, long-term care, and death [4]. Additionally, Hiltunen et al. [2] demonstrated the linear relationship between Fried’s frailty phenotype and OFr score. The relationship between OFr and physical activity was investigated, but its association with systemic health, such as reduced bone mineral density (BMD) or renal function, remains unclear.

OFr is caused not only by muscle weakness related to mastication and swallowing but also by decreased occlusal force and chewing efficiency due to dental caries and periodontal disease [1,5,6]. Osteoporosisis is considered a risk factor for periodontal disease because it causes alveolar bone resorption [7,8]. Alternatively, only limited information is currently available on the association between BMD and OFr [9]. Our previous findings revealed an association between the reduced intake of mineral-containing foods due to OFr and decreased BMD [9]. Although little information is available on BMD and OFr, it has not yet been investigated whether one factor modifies the other, or whether both factors have a reciprocal adverse association.

The relationship between oral and systemic health has revealed that periodontal disease affects chronic kidney disease (CKD) [10,11], in addition to OFr decreasing daily living activities. Renal function, bacteremia and the cytokine-mediated complex pathogenesis of periodontal disease have been implicated in CKD [10,11]. A cross-sectional study by Kosaka et al. [12] reported that CKD severity was not associated with declined chewing or swallowing function. Although Kosaka et al. [12] evaluated tongue and lip motor, masticatory, and swallowing functions as individual factors with CKD, it is necessary to focus on the fact that these factors are not the same as OFr, which evaluates the overall decline of various oral functions. Furthermore, few studies have investigated the relationship between OFr and renal function. Therefore, the relationship between OFr and renal function, including in individuals without CKD, warrants further investigation.

Although only a few epidemiological studies have been conducted on the relationship between OFr and BMD, or between OFr and renal function, the combination of these factors is not yet well known. OFr, accompanied by decreased BMD, a risk factor for periodontal disease, was hypothesized to reduce renal function. Therefore, this cross-sectional study examined the association of OFr with decreased BMD and renal function in residents of Shika town, Ishikawa Prefecture, Japan aged ≥ 40 years.

## 2. Materials and Methods

### 2.1. Study Design, Setting, and Participants

This study was conducted as part of the Shika study with residents of Shika town. Shika town had 20,845 residents, of whom 14937 were aged ≥40 years as of February 2018 [13]. The Shika study is a community-based epidemiological study using data such as questionnaires and medical information [14,15,16]. This study included 444 residents aged 40 years and older in four model districts (Tsuchida, Horimatsu, Togi, and Higashi Masuho) of Shika town who agreed to participate. Of the 444 individuals, 434 had a”number of teeth” assessment and a steo-sono assessment index (the OSI). Eight individuals with OSI outside the range of 1.5–3.5 were excluded from the study, as well as 26 individuals without estimated glomerular filtration rate (eGFR) measurements. Figure 1 shows the participants’ selection criteria. We performed a statistical analysis on 400 individuals. This study was approved by the Ethics Committee of Kanazawa University (No. 1491) following Helsinki Declaration guidelines.

### 2.2. Data Sources and Variables

Health survey data were gathered from the Shika town residents from November 2017 to February 2018. Drinking status (1: non-drinking or drinking less than once a month, 2: drinking at least once a month), smoking status (1: non-smoker or ex-smoker, 2: current smoker), diabetes (1: no; 2: yes), and osteoporosis (1: no; 2: yes) were investigated using a self-reported questionnaire. Body mass index (BMI) was obtained from the Shika study’s medical checkup data.

The OFr assessment components were the Kihon Checklist (KCL) [17,18], the number of teeth, and brushing habits. The KCL is a self-administered checklist for overall health. The items related to oral health consist of the following domains: eating (eating hard foods is more difficult than 6 months ago or hard foods are cut into small pieces), swallowing (sometimes coughing on tea or soup or coughing while eating), and oral dryness (concerned about the dry mouth or have difficulty swallowing because of dry mouth). Previous studies have demonstrated the reliability [19] and validity [20,21] of KCL. The number of teeth, excluding removable dentures, bridge pontics, or dental implants, was assessed by well-trained dentists. The evaluation points for each item were as follows: eating domain (no: 0 points, yes: 2 points), swallowing domain (no: 0 points, yes: 2 points), oral dryness domain (no: 0 points, yes: 1 point), the number of teeth (≥20 teeth: 0 points, <20 teeth: 1 point), and brushing habit (at least twice a day brushing: 0 points, less than twice a day brushing: 1 point). The sum of each point was considered the OFr total score. Additionally, we defined a score of ≥4 as OFr.

The BMD was measured by OSI utilizing a quantitative ultrasound system (AOS-100NW-B, Hitachi Aloka Medical, Tokyo, Japan). The measurement principle is that the ultrasonic signal emitted from the transducer penetrates the right calcaneus and is captured by the other transducer and converted into digital data [22]. The OSI correlates with BMD measured by dual-energy X-ray absorptiometry [23]. The OSI was calculated as follows:OSI = transmission index × speed of sound^2^.

The estimated glomerular filtration rate (eGFR) was calculated as follows to evaluate renal function in this study [24]:eGFR (mL/min/1.73 m^2^) = 194 × serum creatinine^−1.094^ × age^−0.287^ (if female, × 0.739).

Serum creatinine concentrations were measured using an enzymatic method.

### 2.3. Statistical Methods

Participants were classified into the non-OFr group (total scores of 0–2) and the OFr group (total scores of ≥3). Participants were categorized into the eGFR < 60 and the eGFR ≥ 60 groups. Statistical software was IBM Statistical Package for the Social Sciences version 26 for Windows (IBM, Armonk, NY, USA). For the two groups comparison, Student’s *t*-test was used for the mean ± SD variables, and the chi-square test for the *n* (%) categories. A two-way analysis of covariance (ANCOVA) adjusted for age, BMI, drinking status, and smoking status was performed to examine the main effects and interactions between the two OFr groups and two eGFR groups on OSI. A multiple logistic regression analysis, with eGFR as the dependent variable, OSI as the independent variable, and stratified by OFr, was used to confirm the two-way ANCOVA results. Variable selection was based on the forced input method. The significance level was set to 5%.

### 2.4. Sample Size

We used the free software, G-power, to calculate the sample size. In the t-test for two independent groups, effect size, α error probability, and power were set 0.3, 0.05, and 0.8, respectively. The total sample size and actual power were found to be 352 and 0.801. In the F-test for ANCOVA, effect size, alpha error probability, power, number of covariates, and number of groups were set 0.25, 0.05, 0.95, 4, and 4, respectively. The total sample size and actual power were 279 and 0.950. For the Z-tests for logistic regression, tails, odds ratio, mull hypothesis, alpha error probability, power, X distribution, X parm π were set to two, 2.5, 0.20, 0.05, 0.95, binomial, and 0.5, respectively. The total sample size and actual power were found to be 319 and 0.950, respectively. Therefore, the sample size of this study was confirmed to be sufficient.

## 3. Results

### 3.1. Participant Characteristics

Table 1 shows the participant characteristics. Of the 400 participants, 192 were male and 208 were female. The participants’ mean (standard deviation [SD]) age was 60.5 (9.8) years for males, which was not significantly different from that of 60.6 (10.7) years for females. BMI (*p* < 0.001), OSI (*p* < 0.001), and the proportion of drinking status (*p* < 0.001) and smoking status (*p* < 0.001) were significantly higher in males than in females. Alternatively, osteoporosis (*p* = 0.049) and more than twice a day brushing (*p* < 0.001) were significantly more frequent among females. The OFr total scores were 2.0 (1.6) and 1.8 (1.8) and eGFR was 83.7 (17.5) and 85.3 (17.3) for males and females, respectively, with no significant difference in sex.

### 3.2. Comparison of Two OFr Groups

Table 2 shows the comparison of two OFr groups. The mean age of 66.4 (9.3) in the OFr group was significantly older than that of 59.2 (10.0) in the non-OFr group (*p* < 0.001). In the OFr component, the proportion of eating (*p* < 0.001), swallowing (*p* < 0.001), and oral dryness (*p* < 0.001) domains and the OFr total score (*p* < 0.001) were significantly higher in the OFr group than in the non-OFr group. The number of teeth (*p* < 0.001) and the proportion of those brushing more than twice a day (*p* < 0.001) were significantly higher in the non-OFr group than in the OFr group. eGFR was significantly lower in the OFr group than in the non-OFr group (*p* < 0.001).

### 3.3. Comparison of Two eGFR Groups

Table 3 shows a comparison of two eGFR groups. The mean age of 73.0 (7.0) in the eGFR <60 group was significantly older than that of 59.8 (10.0) in the eGFR ≥ 60 group (*p* < 0.001). BIM (*p* < 0.001), and the proportion of male sex (*p* < 0.001), drinking status (*p* < 0.001), diabetes (*p* = 0.021), and osteoporosis (*p* = 0.016) were significantly higher in the eGFR <60 group than that in the eGFR ≥ 60 group. In the OFr component, the proportion of swallowing domain (*p* = 0.023) and brushing more than twice a day (*p* = 0.001) and the OFr total score (*p* = 0.018) were significantly higher in the eGFR <60 group than that in the eGFR ≥ 60 group. The number of teeth (*p* = 0.021) was significantly higher in the eGFR ≥ 60 group than in the eGFR <60 group. eGFR was significantly lower in the eGFR <60 group than that in the eGFR ≥ 60 group (*p* < 0.001).

### 3.4. Main Effects and Interactions between eGFR and OFr Groups on OSI

The group with an eGFR of ≥60 was subdivided into two groups based on OFr, including 307 and 70 participants in the non-OFr and OFr groups, respectively. The group with an eGFR of <60 was subdivided into two groups based on OFr, including 16 and 7 participants in the non-OFr and OFr groups, respectively (Table 4). A two-way ANCOVA was used to examine the main effects and interactions between eGFR and OFr on OSI after adjusting for age, BMI, and smoking and drinking status. OSI showed a significant main effect in the two OFr groups (*p* = 0.029). Additionally, OSI showed a significant interaction between the two eGFR groups and the two OFr groups (*p* = 0.002). Multiple comparisons, using the post hoc Bonferroni analysis, revealed a significantly lower OSI in the OFr group than in the non-OFr group in the group with an eGFR of <60 (*p* = 0.006), but not in the group with an eGFR of ≥60 (Figure 2).

### 3.5. Relationship between OSI and eGFR Stratified by OFr

Table 5 shows the results of the multiple logistic regression analysis. The OSI (β: −7.582; *p* = 0.006, odds ratio: 0.001, 95% CI: 0.000, 0.110) was a significant independent variable in the model adjusted for age, BMI, and drinking and smoking status in the OFr group, but not in the non-OFr group. Therefore, lower BMD is associated with lower renal function only when accompanied by OFr.

## 4. Discussion

Two-way ANCOVA in the present study showed that OSI was significantly lower in the OFr group than in the non-OFr group in the eGFR < 60 groups, whereas OSI was not different between the OFr and non-OFr groups in the eGFR ≥ 60 groups. These results were confirmed by multiple logistic regression analysis stratified by OFr.

A consensus has not yet been reached, although the definition of OFr has been discussed worldwide [1,2,3,4]. Minakuchi et al. [25] placed OFr between good oral health and oral hypofunction with symptoms, such as “decreased articulation, slight choking/spillage while eating, and increase in unchewable foods,” and proposed that three or more of the following seven conditions are associated with oral hypofunction meaning that the oral function is worse than OFr: decreased masticatory function, reduced occlusal force, decreased tongue pressure, decreased tongue–lip motor function, swallowing deterioration, oral dryness, and poor oral hygiene. Alternatively, Hiltunen et al. [2] used the following six items to assess OFr: salivation as normal or dry mouth, the presence of food residue, inability to keep the mouth open during the examination, unclear speech, pureed or soft food diet, and expression of pain during oral examination. An assessment discrepancy was found between oral hypofunction, which is one level worse than OFr as mentioned by Minakuchi et al. [25], and OFr as described by Hiltunen et al. [2], although the symptoms are the same. The number of items assessed in OFr ranged from three by Ramsay et al. [3] to 16 by Tanaka et al. [4]. We adopted an assessment method that includes the three KCL items, in addition to the number of teeth and the frequency of daily tooth brushing, to satisfy the “decreased articulation, slight choking/spillage while eating, and increase in unchewable foods” criterion, although OFr assessment has no gold standard. A study investigating OFr screening items by Nomura et al. reported that the number of remaining teeth and brushing teeth at least twice a day have been valuable in OFr assessment [26]. Additionally, a systematic review of OFr and its determinations by Dibello et al. [27] indicated that tooth loss could be related to infectious disease due to poor oral care. Moreover, a cross-sectional study by Niesten et al. [28] reported that lower brushing frequency since the onset of care-dependency is related to specific frailty-related factors in a care-dependent older population. The frequency of tooth brushing was included in our OFr assessment because plaque control is an effective strategy for preventing tooth loss [29]. Our results comparing the non-OFr and OFr groups showed that all components adopted for the OFr total score showed significant differences in their values, which suggests that it was an appropriate composition. However, the OFr definition and its evaluation items need further discussion to reach a consensus.

Regarding the masticatory domain, a cross-sectional study of the older population demonstrated that a small number of teeth was associated with the decreased masticatory ability [30]. A cross-sectional study of independently living people aged 60–84 years demonstrated that declines in the number of residual teeth, occlusal force, and salivary flow were associated with a masticatory performance reduction [31]. Our results revealed a mean number of teeth of 22.1 ± 7.3 in the non-OFr group, whereas 12.1 ± 8.8 in the OFr group. A decreased occlusal force and chewing efficiency due to dental caries and periodontal disease is considered one of the possible reasons for OFr pathogenesis [1]. Dental caries and periodontal diseases mainly cause tooth loss [32]. Regarding the adverse effects of periodontal disease (the most common cause of OFr) on renal function, bacteremia due to periodontal lesions affect CKD through complex pathogenesis, such as renal fibrosis, endothelial dysfunction, and intestinal flora changes [11]. A systematic review revealed that the probability of having CKD was 60% higher in patients with periodontitis, suggesting the strong potential of periodontal disease to adversely affect renal function [10]. Our results revealed that decreased renal function in the OFr group with decreased BMD, but not in the non-OFr group, is possibly caused by periodontal diseases, in addition to the change in food preferences due to decreased masticatory function. However, very few studies have directly evaluated the effect of OFr with periodontal disease on renal function [12], with more investigations required in the future.

Regarding the swallowing domain, a cross-sectional study by Pinto et al. [33] evaluated swallowing function using videofluoroscopy in 20 hospitalized chronic renal failure patients, and revealed laryngeal penetration and tracheal aspiration in 30% of participants. Additionally, a cross-sectional study by Kosaka et al. [12] evaluated the relationship between renal function, frailty, and OFr and revealed a significant association between eGFR and oral diadochokinesis in a multivariate analysis. Moreover, a case-control study by Oyetola et al. [34] and a comparative study by Vesterinen et al. [35] revealed that patients with CKD have decreased salivary production and complain of xerostomia, respectively. Therefore, the decreased saliva production associated with decreased renal function may affect swallowing capacity due to impaired food lump formation and pharyngeal passing. Our results in the two OFr groups showed significantly lower eGFR values in the OFr group, which seems to support previous findings that poor swallowing is related to renal function. However, masticatory and swallowing functions work together [36], thus clarifying that the relationship between OFr and renal function is necessary to investigate the causal relationship between the decreased masticatory function, and decreased salivary function affecting the swallowing function, or between the decreased renal function and the decreased salivary function affecting the swallowing function.

It is known that reduced BMD results from impaired renal function, which regulates the vitamin D hormonal system [37]. A cohort study of participants aged 65 years and older by Chen et al. [38] reported a relationship between early renal function decline and fracture risk. In contrast, our results showed an association between osteoporosis and eGFR, but no significant difference in OSI between the two eGFR groups. Since our target population was a community of residents over 40 years of age, it could be considered different from studies of older adults with potentially impaired kidney function or patients with severe CKD. Alternatively, in the two eGFR groups comparison, the OFr total score was significantly higher in the eGFR <60 group than in the eGFR ≥60 group. Therefore, our results suggest that the decreased renal function seems more likely to be related to OFr rather than to OSI. Furthermore, our two-way ANCOVA showed that OSI was significantly lower in the OFr group than in the non-OFr group in the eGFR < 60 groups, whereas OSI was not different between the OFr and non-OFr groups in the eGFR ≥ 60 groups. Thus, OFr with decreased BMD seems to be associated with declined renal function, whereas such a relationship is not found without a decrease in BMD. As a possible mechanism, since lower BMD is a risk factor for periodontal disease [7,8], it is suggested that OFr involving periodontal disease seems to be related to the decline in renal function. Furthermore, multiple logistic regression analysis showed that a decreased OSI was an independent variable significantly associated with a reduced eGFR only when accompanied by OFr. Therefore, the clinical significance of this study is that if OFr with decreased BMD is considered to be the cause of reduced renal function, in addition to treatment for decreased BMD, approaches to OFr such as periodontal treatment, eating and swallowing therapy, oral dryness treatment, and oral hygiene instruction would help improve renal function.

The limitation of this cross-sectional study was, first, the impossible elucidation of the causal relationship among the OSI, OFr, and eGFR. Second, OSI does not directly evaluate BMD. Third, OFr was assessed using a checklist, which may lack objectivity. Fourth, the cause for tooth loss was not elucidated. Fifth, diabetes and osteoporosis were self-reported, which may differ from the actual diagnosis. Finally, this study did not analyze functional teeth, nutrient intake, exercise habits, or other systemic diseases.

## 5. Conclusions

Our results revealed that two-way ANCOVA in the present study showed that OSI was significantly negatively associated with eGFR < 60 in the OFr group but not in the non-OFr group. These results were also confirmed by multiple logistic regression analysis stratified by OFr. Therefore, lower BMD seems to be associated with lower renal function only when accompanied by OFr. Further longitudinal studies are needed to confirm these results.

## Figures and Tables

**Figure 1 healthcare-11-00314-f001:**
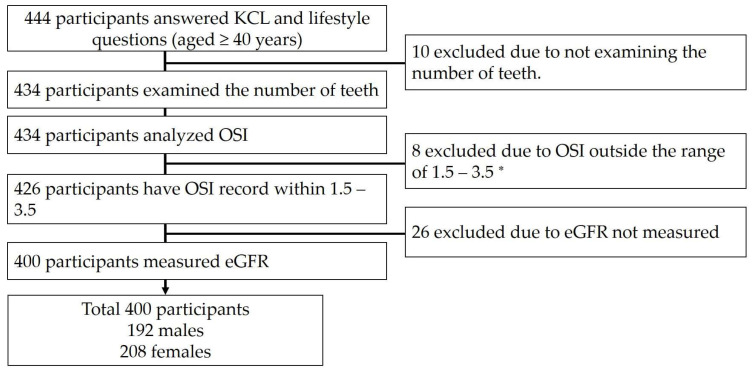
Participant’s selection criteria. * This range was defined from the mean ± 2 SD. Abbreviations: eGFR, estimated glomerular filtration rate; KCL, the Kihon checklist, OSI, the osteo-sono assessment index.

**Figure 2 healthcare-11-00314-f002:**
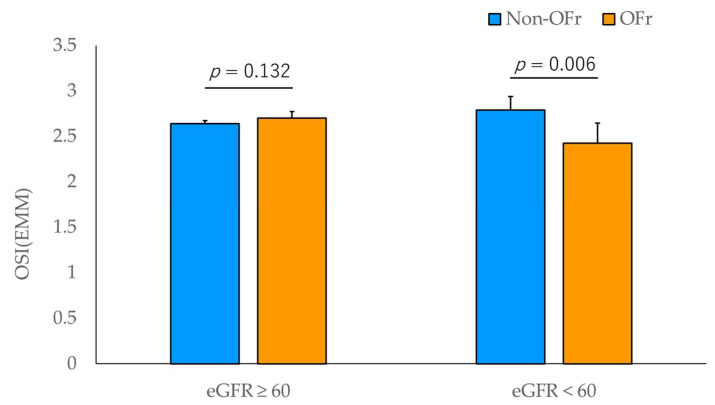
Post hoc Bonferroni analysis. Adjusted for age = 60.56, BMI = 23.33, drinking status = 1.52, and smoking status = 1.16. Error bar: 95% CI. Abbreviations: BMI, body mass index; CI, confidence interval; eGFR, estimated glomerular filtration rate; EMM, estimated marginal means; OFr, oral frailty; OSI, osteo-sono assessment index.

**Table 1 healthcare-11-00314-t001:** Participant characteristics.

	Male (*n* = 192)	Female (*n* = 208)	*p*-Value *
Age, years, mean (SD)	60.5 (9.8)	60.6 (10.7)	0.973
BMI, kg/m^2,^ mean (SD)	24.1 (3.0)	22.6 (3.2)	**<0.001**
Drinking status, *n* (%)	148 (77.1%)	58 (27.9%)	**<0.001**
Smoking status, *n* (%)	51 (26.6%)	11 (5.3%)	**<0.001**
Diabetes, *n* (%)	12 (6.3%)	3 (1.4%)	0.057
Osteoporosis, *n* (%)	1 (0.5%)	6 (2.9%)	**0.049**
OFr			
Eating domain, *n* (%)	51 (26.6%)	55 (26.4%)	1.000
Swallowing domain, *n* (%)	39 (20.3%)	49 (23.6%)	0.470
Oral dryness domain, *n* (%)	39 (20.3%)	45 (21.6%)	0.806
Number of teeth, mean (SD)	20.0 (8.3)	20.4 (8.8)	0.580
More than twice a day brushing, *n* (%)	29 (15.1%)	77 (37.0%)	**<0.001**
OFr total score, mean (SD)	2.0 (1.6)	1.8 (1.8)	0.374
OSI, mean (SD)	2.8 (0.3)	2.5 (0.3)	**<0.001**
eGFR (mL/min/1.73 m^2^), mean (SD)	83.7 (17.5)	85.5 (17.3)	0.294

* *p*-values were calculated using Student’s *t*-tests and Chi-square test for continuous and categorical variables, respectively (*p*-values of <0.05 are highlighted in bold). Abbreviations: BMI, body mass index; eGFR, estimated glomerular filtration rate; OFr, oral frailty; OSI, osteo-sono assessment index; SD, standard deviation.

**Table 2 healthcare-11-00314-t002:** Comparison of the two OFr groups.

	Non-OFr (*n* = 323)	OFr (*n* = 77)	*p*-Value *
Age, years, mean (SD)	59.2 (10.0)	66.4 (9.3)	**<0.001**
Sex (Male), *n* (%)	155 (48.0%)	37 (48.1%)	0.992
BMI, kg/m^2^, mean (SD)	23.2 (3.2)	23.7 (3.1)	0.203
Drinking status, *n* (%)	173 (53.6%)	33 (42.9%)	0.091
Smoking status, *n* (%)	49 (15.2%)	13 (16.9%)	0.709
Diabetes, *n* (%)	12 (3.7%)	3 (3.9%)	0.976
Osteoporosis, *n* (%)	5 (1.5%)	2 (2.6%)	0.580
OFr			
Eating domain, *n* (%)	46 (14.2%)	60 (77.9%)	**<0.001**
Swallowing domain, *n* (%)	44 (13.6%)	44 (57.1%)	**<0.001**
Oral dryness domain, *n* (%)	46 (14.2%)	38 (49.4%)	**<0.001**
Number of teeth, mean (SD)	22.1 (7.3)	12.1 (8.8)	**<0.001**
More than twice a day brushing, *n* (%)	227 (70.3%)	25 (32.5%)	**<0.001**
OFr total score, mean (SD)	1.2 (1.1)	4.6 (0.9)	**<0.001**
OSI, mean (SD)	2.7 (0.3)	2.6 (0.4)	0.337
eGFR (mL/min/1.73 m^2^), mean (SD)	86.1 (16.8)	78.4 (18.3)	**<0.001**

* *p*-values were calculated using Student’s *t*-tests and Chi-square test for continuous and categorical variables, respectively (*p*-values of <0.05 are highlighted in bold). Abbreviations: BMI, body mass index; eGFR, estimated glomerular filtration rate; OFr, oral frailty; OSI, osteo-sono assessment index; SD, standard deviation.

**Table 3 healthcare-11-00314-t003:** Comparison of the two eGFR groups.

	eGFR ≥ 60(*n* = 377)		eGFR <60 (*n* = 23)		*p*-Value *
Age, years, mean (SD)	59.8	10.0	73.0	7.0	**<0.001**
Sex (Male), *n* (%)	179.0	47.5	13.0	56.5	**<0.001**
BMI, kg/m^2^, mean (SD)	23.2	3.1	25.9	3.9	**<0.001**
Drinking status, *n* (%)	193	51.2	13	56.5	**<0.001**
Smoking status, *n* (%)	59	15.6	3	13.0	**<0.001**
Diabetes, *n* (%)	12	3.2	3	13.0	**0.021**
Osteoporosis, *n* (%)	4	1.1	3	13.0	**0.016**
OFr					
Eating domain, *n* (%)	95	25.2	11	47.8	0.508
Swallowing domain, *n* (%)	81	21.5	7	**30.4**	**0.023**
Oral dryness domain, *n* (%)	77	20.4	7	30.4	0.192
Number of teeth, mean (SD)	**20.5**	8.4	16.2	9.4	**0.021**
More than twice a day brushing, *n* (%)	237	62.9	15	**65.2**	**0.001**
OFr total score, mean (SD)	1.8	1.7	**2.7**	2.1	**0.018**
OSI, mean (SD)	2.7	0.3	2.6	0.4	0.810
eGFR (mL/min/1.73 m^2^), mean (SD)	86.7	15.5	**50.3**	9.2	**<0.001**

* *p*-values were calculated using Student’s *t*-tests and Chi-square test for continuous and categorical variables, respectively (*p*-values of <0.05 are highlighted in bold). Abbreviations: BMI, body mass index; eGFR, estimated glomerular filtration rate; OFr, oral frailty; OSI, osteo-sono assessment index; SD, standard deviation.

**Table 4 healthcare-11-00314-t004:** Two-way ANCOVA of eGFR and OFr groups on OSI.

	eGFR ≥ 60 (*n* = 377) EMM (95%CI)	eGFR < 60 (*n* = 23) EMM (95%CI)	*p*-Value *		
			Main Effect		Interaction
			eGFR	OFr	eGFR × OFr
Non-OFr	2.64 (2.61, 2.67)	2.79 (2.64, 2.94)	0.381	**0.029**	**0.002**
	(*n* = 307)	(*n* = 16)			
OFr	2.70 (2.63, 2.77)	2.42 (2.20, 2.65)			
	(*n* = 70)	(*n* = 7)			

Dependent variables are OSI. * Two-way ANCOVA (*p*-values of <0.05 are highlighted in bold). Adjusted for age, BMI, and smoking and drinking status. Abbreviations: ANCOVA, analysis of covariance; BMI, body mass index; CI, confidence interval; eGFR, estimated glomerular filtration rate; EMM, estimated marginal means; OFr, oral frailty; OSI, osteo-sono assessment index.

**Table 5 healthcare-11-00314-t005:** Relationship between OSI and eGFR stratified by OFr.

		β	*p*-Value	OR	95%CI (Lower, Upper)
Non-OFr	Age	0.037	0.104	1.037	0.993, 1.084
(*n* = 323)	BMI	0.052	0.508	1.053	0.903, 1.229
	Drinking status	0.283	0.588	1.327	0.477, 3.690
	Smoking status	−2.661	0.153	0.070	0.002, 2.683
	OSI	−1.514	0.064	0.220	0.044, 1.091
OFr	Age	0.085	0.050	1.088	1.000, 1.184
(*n* = 77)	BMI	0.266	0.084	1.304	0.965, 1.764
	Drinking status	0.785	0.498	2.193	0.226, 21.239
	Smoking status	2.221	0.215	9.219	0.275, 309.005
	OSI	−7.582	**0.006**	0.001	0.000, 0.110

Multiple logistic regression analysis. The dependent variable is eGFR. Significant estimates are in bold. Abbreviations: β, coefficient; BMI, body mass index; CI, confidence interval; eGFR, estimated glomerular filtration rate; OFr, oral frailty; OR; odds ratio; OSI, osteo-sono assessment index.

## Data Availability

Data in the present study are available upon request from the corresponding author. Data are not publicly available due to privacy and ethical policies.

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
