# Peer review of "Association between Bone Mineral Density and Oral Frailty on Renal Function: Findings from the Shika Study"

_healthcare, 2023, doi:10.3390/healthcare11030314_

Round 1
Reviewer 1 Report (Previous Reviewer 1)
Thank you for the revision.
Author Response
Reviewer 1
Comment 1
Thank you for the revision.
Response 1
We appreciate the time and effort you have dedicated to providing insightful feedback on ways to strengthen our paper.
Reviewer 2 Report (Previous Reviewer 2)
Thank you for giving me the opportunity to review the revised manuscript titled “Association between Bone Mineral Density and Oral Frailty on Renal Function: Findings from the Shika Study.” The authors re-analyzed the data and I think that this manuscript was improved.
Table 5 shows the multiple regression analysis results stratified by OFr, with eGFR as the dependent variable and the OSI as the independent variable. The OSI (P=0.006, odds ratio: 0.001) was a significant independent variable. I think that the odds ratio was very low, therefore, the clinical relevance should be discussed.
Author Response
Reviewer 2
Comment 1
Thank you for giving me the opportunity to review the revised manuscript titled “Association between Bone Mineral Density and Oral Frailty on Renal Function: Findings from the Shika Study.” The authors re-analyzed the data and I think that this manuscript was improved.
Table 5 shows the multiple regression analysis results stratified by OFr, with eGFR as the dependent variable and the OSI as the independent variable. The OSI (P=0.006, odds ratio: 0.001) was a significant independent variable. I think that the odds ratio was very low, therefore, the clinical relevance should be discussed.
Response 1
We have added the following sentence to the Discussion section:
“Furthermore, multiple logistic regression analysis showed that a decreased OSI was an independent variable significantly associated with a reduced eGFR only in the presence of OFr. Therefore, the clinical significance of this study is that if OFr with decreased BMD is considered to be the cause of reduced renal function, in addition to treatment for decreased BMD, approaches to OFr such as periodontal treatment, eating and swallowing therapy, oral dryness treatment, and oral hygiene instruction seems to be help improve renal function.” (L342-348)
This manuscript is a resubmission of an earlier submission. The following is a list of the peer review reports and author responses from that submission.
Round 1
Reviewer 1 Report
The reviewer really appreciates the efforts of the authors to conduct this study which has good clinical significance. The overall writing of the manuscript looks good. However, there are several facts that need to be justified in order to understand the outcome clearly.
The factors like drinking status, smoking status, and diabetes were calculated in a chronological manner (yes/ no) which was converted as numerical 1/2/3. The reviewer could not figure out how yes/no answers have a mean value and standard deviation as described in table 1. The reviewer’s understanding is that this kind of parameter can be expressed in the form of frequency.
The second concern is the statement “A two-way analysis of covariance (ANCOVA) adjusted for age, sex, drinking status, and smoking status was performed to examine the main effects and interactions between the two BMD groups and two eGFR groups on OFr”. The reviewer’s understanding is two-way ANOVA is indicated when you want to evaluate the influence/ interaction of two factors on a certain value. The author has included four factors in the analysis.
In the same way, the author used a multiple regression assay to check the influence of Two factors which is usually used to check the correlation of multiple factors (more than 3) with one value.
The reviewer's suggestion would be to consult with a good statistician to revise the statistical analysis.
Author Response
Reviewer 1
The reviewer really appreciates the efforts of the authors to conduct this study which has good clinical significance. The overall writing of the manuscript looks good. However, there are several facts that need to be justified in order to understand the outcome clearly.
Comment 1
The factors like drinking status, smoking status, and diabetes were calculated in a chronological manner (yes/ no) which was converted as numerical 1/2/3. The reviewer could not figure out how yes/no answers have a mean value and standard deviation as described in table 1. The reviewer’s understanding is that this kind of parameter can be expressed in the form of frequency.
Response 1
We have presented yes/no questions as n (%) in Tables 1 through 3. To avoid misunderstandings, we have corrected them to mean ± SD / n (%).
Comment 2
The second concern is the statement “A two-way analysis of covariance (ANCOVA) adjusted for age, sex, drinking status, and smoking status was performed to examine the main effects and interactions between the two BMD groups and two eGFR groups on OFr”. The reviewer’s understanding is two-way ANOVA is indicated when you want to evaluate the influence/ interaction of two factors on a certain value. The author has included four factors in the analysis.
Response 2
We do not use four different groups. The table shows two factors as 2 x 2. Specifically, the two eGFR groups are shown as the upper classification and the two BMD groups (OSI in the revised manuscript) as the lower classification.
Comment 3
In the same way, the author used a multiple regression assay to check the influence of Two factors which is usually used to check the correlation of multiple factors (more than 3) with one value.
The reviewer's suggestion would be to consult with a good statistician to revise the statistical analysis.
Response 3
One method of multiple regression analysis uses an interaction term as an independent variable for one dependent variable. However, biological responses are not simply multiplicative; there are thresholds. In other words, they are involved within the range but not outside of that range. Therefore, instead of using an interaction term, we have adopted the method of stratifying the groups to show that the independent variable (OFr total score) is involved in one condition (low OSI) but not in the other (high OSI).
We added Table S1 with the interaction term between OFr total score and OSI as independent variables to supplement the results in Table 5 and amended the results in the revised manuscript as follows:
“This result was also shown to be a significant independent variable in a multiple regression analysis using an interaction term between OFr total score and OSI β: -0.051; 95%CI: -1.174, -0.164; p = 0.010) (Table S1).” (L769-771)

Reviewer 2 Report
Thank you for giving me the opportunity to review the manuscript titled “Associationbetween Bone Mineral Density and Oral Frailty on Renal Function: Findings from the Shika Study.”
This paper demonstrated that oral frailty prevention is an effective strategy for maintaining renal function, particularly in people with low bone mineral density (BMD). Although this is a fascinating new initiative, several major issues must be addressed before this manuscript can be considered suitable for publication.
1. In the Introduction section, I believe that references 10–12 were unrelated to this study. Please verify them.
2. It was perplexing that the authors excluded eight participants with OSI 1.5–3.5 (L99). OSI outside the 1.5–3.5 range?
3. The authors gathered data on systemic diseases, but only diabetes and osteoporosis were included. A self-administered questionnaire was used to investigate the two diseases. This measurement method, in my opinion, is debatable. As a result, the authors should mention it in the Discussion section. Furthermore, no research was conducted into nutrient status, exercise habits, or other systemic diseases associated with BMD and CKD. These factors must be considered.
4. The criteria for judging oral frailty are baffling. The five items were not distinct. Furthermore, I do not understand how brushing times are related to oral frailty. Is brushing one of the healthy habits? Because dental implants work properly, I believe that the authors should examine the number of functional teeth rather than the number of remaining teeth.
5. The authors divided the participants into low and high BMD groups based on the median number of participants. Is this a viable method? It is recommended to use a reference value if one exists.
6. In tables, does BMD denote OSI? I am confused.
7. Due to my comments in no. 4, I did not agree that oral frailty was analyzed by total score in Table 5.
The findings were intriguing, but ambiguous evaluation methods could lead to incorrect results.
Author Response
Reviewer 2
Thank you for giving me the opportunity to review the manuscript titled “Association between Bone Mineral Density and Oral Frailty on Renal Function: Findings from the Shika Study.”
This paper demonstrated that oral frailty prevention is an effective strategy for maintaining renal function, particularly in people with low bone mineral density (BMD). Although this is a fascinating new initiative, several major issues must be addressed before this manuscript can be considered suitable for publication.
Comment 1
- In the Introduction section, I believe that references 10–12 were unrelated to this study. Please verify them.
Response 1
We have removed the following sentence from the Introduction section:
“aspiration pneumonia [10], cardiovascular disease [11], diabetes [12], and”
Comment 2
- It was perplexing that the authors excluded eight participants with OSI 1.5–3.5 (L99). OSI outside the 1.5–3.5 range?
Response 2
We have amended the Study Design, Setting, and Participants section as follows:
“Eight individuals with OSI outside the range of 1.5–3.5 were excluded from the study,” (L108-109)
Comment 3
- The authors gathered data on systemic diseases, but only diabetes and osteoporosis were included. A self-administered questionnaire was used to investigate the two diseases. This measurement method, in my opinion, is debatable. As a result, the authors should mention it in the Discussion section. Furthermore, no research was conducted into nutrient status, exercise habits, or other systemic diseases associated with BMD and CKD. These factors must be considered.
Response 3
We have added the following sentence to the limitation section:
"Fifth, diabetes and osteoporosis were self-reported, which may differ from the actual diagnosis. Finally, this study did not analyze functional teeth, nutrient intake, exercise habits, or other systemic diseases." (L865-867)
Comment 4
- The criteria for judging oral frailty are baffling. The five items were not distinct. Furthermore, I do not understand how brushing times are related to oral frailty. Is brushing one of the healthy habits? Because dental implants work properly, I believe that the authors should examine the number of functional teeth rather than the number of remaining teeth.
Response 4
Even dental implants can lead to peri-implantitis without proper plaque control.
We have added the following sentence to the discussion section regarding the article on the number of remaining teeth and the frequency of tooth brushing related to oral frailty:
“A study investigating OFr screening items by Nomura et al. [26] reported that the number of remaining teeth and brushing teeth at least twice a day have been valuable in the OFr assessment. Additionally, a systematic review of OFr and its determinations by Dibello et al. [27] indicated that tooth loss could be related to infectious disease due to poor oral care. Moreover, a cross-sectional study by Niesten et al. [28] reported that lower brushing frequency since the onset of care-dependency is related to specific frailty-related factors in a care-dependent older population.” (L799-818)
“Finally, this study did not analyze functional teeth, nutrient intake, exercise habits, or other systemic diseases.” (L866-867)
Comment 5
- The authors divided the participants into low and high BMD groups based on the median number of participants. Is this a viable method? It is recommended to use a reference value if one exists.
Response 6
According to the paper (in Japanese) on the AOS100, an OSI measuring instrument, reference values are given for each age group by sex. However, we adopted the Median split because we considered it inappropriate to set individual reference values for each subject for statistical analysis.
https://mol.medicalonline.jp/library/journal/download?GoodsID=ai6ostoe/2005/001301/005&name=0031-0035j&UserID=221.186.161.20&base=jamas_pdf
Comment 6
- In tables, does BMD denote OSI? I am confused.
Response 6
We have amended the BMD to OSI.
Comment 7
- Due to my comments in no. 4, I did not agree that oral frailty was analyzed by total score in Table 5.
The findings were intriguing, but ambiguous evaluation methods could lead to incorrect results.
Response 7
We have shown in Response 4 that number of teeth and frequency of tooth brushing are used to assess oral frailty and that frequency of tooth brushing is also related to systemic frailty. Therefore, we believe that the results in Table 5 are appropriate.

Round 2
Reviewer 1 Report
The revised version did not improve the overall quality of the presentation. The tables became difficult to understand. Despite having a rationale, the writing/ presentation style used by the author needs to be improved prior consider it for publication.
Author Response
Comment 1
The revised version did not improve the overall quality of the presentation. The tables became difficult to understand. Despite having a rationale, the writing/ presentation style used by the author needs to be improved prior consider it for publication.
Response 1
We have modified the tables' expression regarding an article on epidemiological studies previously published in Healthcare.
Additionally, we have made revisions after careful inspection of the Introduction to the Conclusions.

Reviewer 2 Report
Thank you for submitting the revised manuscript.
I can not agree that oral frailty was analyzed by total score in Table 5. In this case, oral frailty should not be discussed in continuous variables
The authors mentioned, "we adopted the Median split because we considered it inappropriate to set individual reference values for each subject for statistical analysis." I think that they can reanalyze.
Author Response
Reviewer 2
Thank you for submitting the revised manuscript.
Comment 1
I can not agree that oral frailty was analyzed by total score in Table 5. In this case, oral frailty should not be discussed in continuous variables
Response 1
We have modified our multivariate analysis in Table 5 (new Table 4) not to use OFr as a continuous variable.
Comment 2
The authors mentioned, "we adopted the Median split because we considered it inappropriate to set individual reference values for each subject for statistical analysis." I think that they can reanalyze.
Response 2
We have removed from the Statistical Methods the description of dividing OSI into two groups by median.
Additionally, we removed the old Table 3, which divided OSI into two groups, and used it only as a continuous variable in the analysis.
